# QC-Bench: What Do Language Models Know About Quantum Computing?

## Abstract

Language models have become practical tools for quantum computing education and research, from summarizing technical papers to explaining theoretical concepts. While existing benchmarks evaluate quantum code generation and circuit design, their understanding of quantum computing concepts has not been systematically measured. QC-Bench addresses this gap with over 6,000 expert-level questions on quantum algorithms, error correction, and security protocols. Evaluating 31 models from OpenAI, Anthropic, Google, and Meta reveals strong performance on established theory but systematic failures on advanced topics like quantum security and recent attack vectors. Human participants scored between 23% and 86%, with experts averaging 74% and all participants averaging 57%. Top-performing models exceeded the expert average, with Claude Sonnet 4 and GPT-5 reaching 88% overall, yet dropping to 76% on security questions. Additional evaluation across question formats and languages reveals variation in model performance, demonstrating that QC-Bench provides a necessary framework for measuring language model reliability in quantum computing contexts.

## 1 Introduction

Quantum computing has progressed significantly from theoretical research to experimental implementations with practical applications. Current quantum systems have rapidly evolved through successive technological breakthroughs from operating with just a few qubits to recently surpassing the 1000-qubit barrier AbuGhanem (2025), enabling exploration of quantum algorithms and protocols that were previously confined to theoretical analysis. This technical advancement drives progress in quantum simulation King et al. (2025); Halimeh et al. (2025); Puig et al. (2025), optimization problems Quinton et al. (2025); Phillipson (2024), and cryptographic applications Sahu & Mazumdar (2024); Ralegankar et al. (2021); Kalaivani et al. (2021). Beyond traditional quantum applications such as quantum simulation and cryptography, recent research explores its potential in finance Innan et al. (2024); Grossi et al. (2022), healthcare Ur Rasool et al. (2023); Flöther (2023), computer vision Li et al. (2020); Afane et al. (2025); ALRikabi et al. (2022), and wireless communication Narottama & Shin (2021); Narottama et al. (2023), among other promising real-world applications.

In parallel, Large Language Models (LLMs) have become sophisticated tools that address complex challenges across many disciplines. These AI systems now approach or exceed human expert performance in areas such as cybersecurity Tihanyi et al. (2024); Afane et al. (2024), medical diagnosis Subedi (2025), and legal reasoning Guha et al. (2023); Kant et al. (2025). As these two fields continue to evolve, their intersection becomes increasingly important for scientific communication, education, and research productivity. Despite significant advances in both domains, we face a critical knowledge gap in evaluating LLMs' understanding of specialized quantum concepts. While extensive benchmarking exists across numerous related domains, including mathematics Gao et al. (2024); Fang et al. (2024), physics Chung et al. (2025), and computer science Song et al. (2024), no standardized frameworks comprehensively assess quantum computing knowledge in these models. This absence is particularly concerning given the field's counterintuitive principles, and rapidly evolving terminology that challenge even domain experts. The complexity of quantum computing concepts, combined with their inherent mathematical abstraction, creates a particularly demanding test case for evaluating the depth of LLMs' specialized knowledge. Without reliable evaluation metrics, LLMs risk spreading plausible but incorrect quantum information to educational and research communities, as

hallucinations, reasoning errors, and factual inaccuracies have been widely documented in similarly complex and technically demanding specialized domains. Orgad et al. (2024); Perković et al. (2024).

This creates an urgent need for robust quantum computing benchmarks as researchers, students, and industry professionals increasingly rely on these models for information and assistance with quantum tasks. The growing adoption of LLMs across academic institutions and quantum technology companies further amplifies the importance of ensuring these systems provide accurate information on this emerging field. To address these challenges, we present the following key contributions:

- We assemble **6,237 questions**: 5,400 multiple-choice questions comprising QC1000 (1,000 entirely human-authored from peer-reviewed literature, with QC500 translated into Spanish and French) and 4,400 human-validated questions filtered from 8,686 candidates, plus 837 format variants (416 true/false, 421 open-ended) for evaluating model performance across question formats.
- We conduct extensive evaluation across **31 models** from leading AI research organizations including OpenAI, Anthropic, Google, Meta, IBM, Microsoft, and DeepSeek, among others. We compare their performance against 43 quantum computing experts and practitioners to establish human baselines and assess how LLMs perform relative to human capabilities.
- We analyze model performance across different question formats and via Spanish and French translations of QC500, revealing significant accuracy declines in the translated sets and consistent sensitivity to question type, with larger drops in Spanish than in French.
- We explore the potential of our dataset for fine-tuning by using a subset of 4,000 questions to enhance the quantum knowledge of five smaller models, demonstrating performance improvements and establishing the benchmark's value beyond evaluation.

## 2 RELATED WORK

Despite significant advancements in both quantum computing and LLMs, their intersection remains surprisingly underexplored. Recent research has begun addressing this gap from different angles. Kashani Kashani (2024) introduced QuantumLLMInstruct (QLMMI), a dataset of over 500,000 instruction-problem pairs covering quantum cryptography, spin chain models, and Trotter-Suzuki decompositions. However, QLMMI's primary purpose is to enable instruction fine-tuning rather than comprehensive evaluation of quantum knowledge. While extensive in size, QLMMI relies entirely on synthetically generated content through a four-stage LLM pipeline. In contrast, QC-Bench offers 1,200 human-authored evaluation questions extracted directly from research literature published over four decades, prioritizing authentic scientific content over synthetic generation. Wang et al. Wang et al. (2024) introduced GroverGPT, an approach to simulating quantum algorithms using LLMs. Their 8-billion-parameter model is fine-tuned to approximate Grover's quantum search algorithm without explicitly representing quantum states. While GroverGPT demonstrates impressive capabilities in predicting specific quantum circuit outputs, it focuses exclusively on a single quantum algorithm rather than evaluating comprehensive knowledge across the quantum computing domain.

Complementary efforts have emerged focusing on quantum code generation and circuit implementation capabilities. Vishwakarma et al. Vishwakarma et al. (2024) developed Qiskit HumanEval, a hand-curated benchmark of over 100 tasks designed to evaluate LLM performance in generating executable quantum code using the Qiskit SDK, complete with canonical solutions and comprehensive test cases. Guo et al. Guo et al. (2025) introduced QuanBench, which evaluates quantum code generation across 44 programming tasks using both functional correctness (Pass@K) and quantum semantic equivalence (Process Fidelity) metrics, finding that current LLMs achieve below 40% overall accuracy with frequent semantic errors including outdated API usage and incorrect algorithm logic. Yang et al. Yang et al. (2024) presented QCircuitNet, a large-scale hierarchical dataset for quantum algorithm design containing 120,290 data points with automatic syntax and semantic verification functions. At a lower abstraction level, Li et al. Li et al. (2023) developed QASMBench, a benchmark suite of low-level OpenQASM programs for evaluating NISQ devices and simulators. While these works provide valuable resources for assessing programming proficiency and implementation capabilities at various levels of quantum software development, they primarily target coding skills rather than evaluating deep conceptual understanding of quantum computing principles, algorithmic theory, or the ability to reason about quantum phenomena. QC-Bench addresses this by

evaluating theoretical knowledge and conceptual understanding across quantum computing topics, from foundational algorithmic principles to advanced security protocols and attack vectors.

## 3 QC-BENCH DATASET

We constructed the QC-Bench dataset to evaluate quantum computing knowledge in LLMs across a wide range of topics and difficulty levels. To ensure comprehensive coverage and relevance, our team reviewed over 200 peer-reviewed research papers, preprints, and academic resources. From these sources, questions were directly selected to reflect both foundational knowledge and current advancements in the field. The dataset comprises QC1000, containing 1000 questions manually extracted from quantum computing literature, with QC500 as a 500-question subset selected for multilingual evaluation. To address concerns about model memorization, none of these questions are reproduced verbatim from source materials; instead, we extracted core concepts and reformulated them into original questions. This approach ensures that performance reflects genuine understanding rather than memorization of published text. After refining and validating this content, the benchmarks were finalized. The QC500 subset was translated into Spanish and French to evaluate LLM performance in languages other than English.

To expand our benchmark, Gemini 2.0 Flash, Gemini 1.5 Pro, GPT-4.0, and Claude 3.7 Sonnet were employed to extract additional relevant questions from the selected papers. Different prompt engineering techniques were tested to optimize question generation quality. While zero-shot prompting produced acceptable results, few-shot prompting with five carefully selected examples from the existing subsets significantly improved the relevance and technical accuracy of generated questions. This approach generated 8,686 candidate questions, subsequently filtered to remove low-quality or redundant items. The final selection included an additional 4,400 high-quality questions, bringing the total benchmark size to 5,400 multiple-choice questions. To evaluate model performance across different question formats, the benchmark was supplemented with 416 true/false questions and 421 open-ended questions. Figure 1 illustrates the distribution of these question types across different topics. Given the interconnected nature of quantum computing, some concepts naturally appear across multiple categories; for example, noise characterization relates to both error correction and hardware-level circuit design. The multiple-choice format enables precise evaluation of factual recall and conceptual understanding, while true/false questions assess binary comprehension, and open-ended questions evaluate explanatory capabilities.

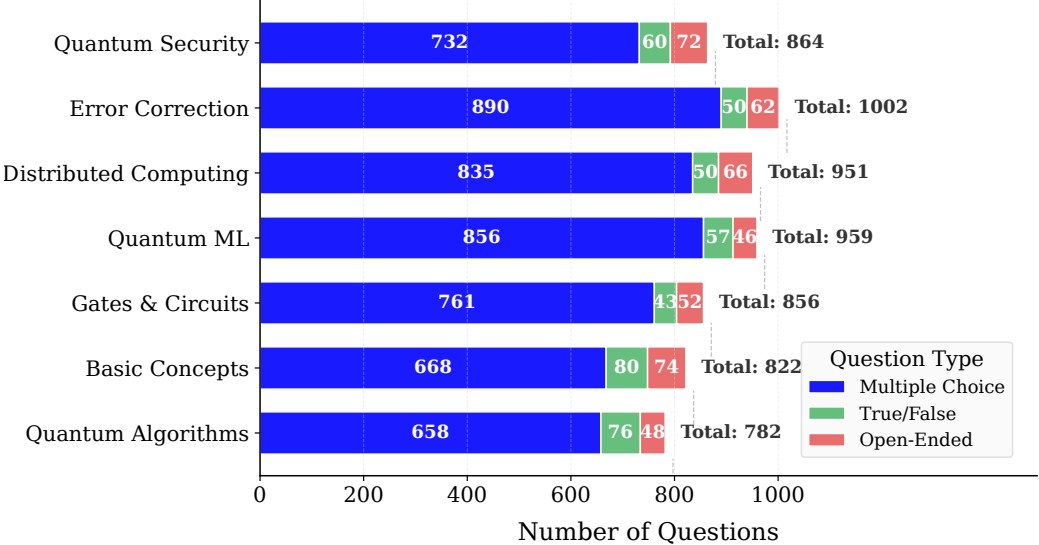

Figure 1: Breakdown of benchmark question topics and their internal composition by question type. Each horizontal bar shows the total number of questions per topic. We intentionally included a larger share of multiple choice items to enable standardized automated evaluation, whereas true/false items offer limited challenge and open-ended questions require manual scoring.

# 4 EXPERIMENTS

We evaluated 31 LLMs using a consistent benchmarking pipeline. Closed-source models, including GPT-5, GPT-4o (standard, mini), Claude (Sonnet 4, Sonnet 3.7, Haiku 3.5), and Gemini (1.5 Pro, 2.0 Flash), were accessed through their official APIs in Google Colab environments. Open-access models such as LLaMA3 (1B, 8B, 70B), LLaMA2 (13B), Phi (2.7B, 3.8B, 14.7B), Mistral (7B, 24B), Qwen1.5 (2.7B, MoE-A2.7B), Zephyr, DeepSeek, Gemma, Granite, and GPT-J were deployed using Hugging Face's Transformers library on a cluster equipped with two Tesla V100 GPUs (32GB each) using FP16 inference. For several larger models, including llama-3 (70b, 70b-versatile), and Gemma-9B, we used Groq's API instead of Hugging Face's Transformers library for a faster and more efficient evaluation. All models were configured with a temperature setting of 1 to balance deterministic responses with reasonable diversity in answer generation.

For experiment preparation, all benchmark questions were structured in JSON format for efficient processing and consistent evaluation across different model architectures. We developed standardized prompting templates for each question type to ensure fair comparison between models. This data preparation approach facilitated automated evaluation pipelines and ensured comparable results despite the diversity of model implementations and access methods. The benchmark includes multiple-choice, true/false, and open-ended formats, with multilingual versions available for a subset of questions. Key findings from these experiments are presented in the following subsections, with complete results and detailed analyses available in the appendix.

| LLM Model | Provider | Size | Access | Q500 | Q1000 | Q5400 |
|---|---|---|---|---|---|---|
| Claude Sonnet 4 | Anthropic | N/A | Anthropic API | 91.80 | 89.90 | 88.55 |
| GPT-5 | OpenAI | N/A | OpenAI API | 91.40 | 90.90 | 88.46 |
| GPT-4o | OpenAI | N/A | OpenAI API | 88.20 | 86.30 | 88.07 |
| Claude Sonnet 3.7 | Anthropic | N/A | Anthropic API | 92.40 | 84.70 | 87.98 |
| GPT-4.1 mini | OpenAI | N/A | OpenAI API | 87.20 | 82.30 | 86.42 |
| Gemini 2.0 Flash | Google | N/A | Google API | 82.40 | 84.60 | 84.44 |
| Gemini 1.5 Pro | Google | N/A | Google API | 80.20 | 84.80 | 83.92 |
| GPT-4o-mini | OpenAI | N/A | OpenAI API | 80.00 | 81.90 | 83.85 |
| llama-3.3-70b-versatile | Meta | 70B | Groq API | 81.40 | 82.00 | 82.07 |
| Phi-4-reasoning-plus | Microsoft | 14.7B | HuggingFace | 87.00 | 89.30 | 81.74 |
| Claude Haiku 3.5 | Anthropic | N/A | Anthropic API | 80.00 | 82.80 | 80.44 |
| granite-3.3-8b-instruct | IBM | 8.17B | HuggingFace | 84.20 | 81.10 | 76.07 |
| Llama-3.1-8B-Instruct | Meta | 8.03B | HuggingFace | 73.80 | 78.40 | 75.75 |
| Phi-4-reasoning | Microsoft | 14.7B | HuggingFace | 81.00 | 80.20 | 75.59 |
| GPT-4.1 nano | OpenAI | N/A | OpenAI API | 86.00 | 86.20 | 74.58 |
| zephyr-7b-beta | Hugging Face | 7.24B | HuggingFace | 84.00 | 83.00 | 73.70 |
| DeepSeek-R1-Dist-Llama-8B | DeepSeek | 8.03B | HuggingFace | 78.00 | 85.20 | 73.62 |
| gemma2-9b-it | Google | 9B | Groq API | 84.60 | 86.40 | 73.55 |
| DeepSeek-R1-Dist-Qwen-7B | DeepSeek | 7.62B | HuggingFace | 78.20 | 86.90 | 72.51 |
| Llama-3.1-8B | Meta | 8B | HuggingFace | 81.00 | 79.50 | 72.51 |
| Mistral-7B-Instruct-v0.3 | Mistral AI | 7.25B | HuggingFace | 82.00 | 80.90 | 72.43 |
| Phi-4-mini-reasoning | Microsoft | 3.84B | HuggingFace | 72.00 | 69.10 | 72.40 |
| llama3-70b | Meta | 70B | Groq API | 84.20 | 82.30 | 71.85 |
| Llama-2-13b-chat-hf | Meta | 13B | HuggingFace | 86.40 | 89.10 | 71.79 |
| Llama-3.2-1B-Instruct | Meta | 1.24B | HuggingFace | 82.20 | 86.00 | 71.55 |
| gemma-7b | Google | 7B | HuggingFace | 72.80 | 74.30 | 69.70 |
| phi-2 | Microsoft | 2.7B | HuggingFace | 81.20 | 78.50 | 67.85 |
| gemma-2-2b-it | Google | 2.61B | HuggingFace | 74.20 | 60.30 | 62.29 |
| Qwen1.5-MoE-A2.7B | Qwen | 14.3B | HuggingFace | 74.00 | 61.70 | 60.74 |
| EleutherAI/gpt-j-6b | EleutherAI | 6B | HuggingFace | 72.00 | 60.90 | 50.14 |
| dolly-v1-6b | Databricks | 6B | HuggingFace | 36.80 | 34.30 | 48.29 |

Table 1: Evaluated language models with provider, size, access method, and accuracy on QC500, QC1000, and QC5400. Rows shaded in green mark the highest performing models overall, and rows shaded in light blue mark the best performing open-source models.

## 4.1 COMPREHENSIVE MODEL EVALUATION ON CORE BENCHMARK AND ACROSS TOPICS

Table 1 details the characteristics of each evaluated model and summarizes performance across the three benchmark subsets. Results from these experiments demonstrate that increasing dataset size from 500 to 5,400 questions does not substantially impact relative model performance.

| Model | Error Correction | Quantum Algorithms | Quantum Security |
|---|---|---|---|
| Claude Sonnet 4 | **92.81** | 81.76 | **76.09** |
| GPT-5 | 92.13 | **82.30** | 75.82 |
| GPT-4o | 92.02 | 79.18 | 75.68 |
| Claude Sonnet 3.7 | 91.12 | 79.03 | 75.00 |
| GPT-4.1 mini | 90.67 | 77.51 | 74.73 |
| Gemini 1.5 Pro | 88.99 | 77.05 | 73.36 |
| Gemini 2.0 Flash | 89.66 | 76.14 | 73.09 |
| GPT-4o-mini | 92.02 | 84.27 | 72.95 |
| Claude Haiku 3.5 | 83.71 | 74.16 | 71.17 |
| llama3-70b | 82.13 | 74.01 | 70.63 |
| Phi-4-reasoning-plus | 81.01 | 82.08 | 69.95 |
| llama-3.3-70b-versatile | 79.89 | 79.39 | 69.95 |
| GPT-4.1 nano | 79.10 | 69.89 | 68.99 |
| granite-3.3-8b-instruct | 77.64 | 70.20 | 65.16 |
| Llama-3.1-8B-Instruct | 77.19 | 67.92 | 64.62 |
| zephyr-7b-beta | 75.39 | 68.28 | 64.07 |
| gemma2-9b-it | 73.15 | 79.75 | 61.61 |
| DeepSeek-R1-Distill-Llama-8B | 73.15 | 65.23 | 60.38 |
| DeepSeek-R1-Distill-Qwen-7B | 72.25 | 73.66 | 58.33 |
| Llama-3.1-8B | 68.88 | 60.75 | 55.87 |
| Phi-4-reasoning | 67.30 | 75.63 | 56.46 |
| Mistral-7B-Instruct-v0.3 | 66.85 | 74.55 | 51.91 |
| Llama-2-13b-chat-hf | 65.96 | 52.33 | 51.78 |
| Llama-3.2-1B-Instruct | 64.38 | 41.58 | 51.09 |
| Phi-4-mini-reasoning | 63.93 | 59.4 | 50.41 |
| gemma-7b | 62.70 | 53.76 | 48.22 |
| Qwen1.5-MoE-A2.7B | 47.30 | 38.53 | 46.45 |
| phi-2 | 58.20 | 37.63 | 43.72 |
| gemma-2-2b-it | 53.93 | 27.24 | 40.16 |
| EleutherAI/gpt-j-6b | 36.63 | 24.55 | 38.52 |
| dolly-v1-6b | 25.84 | 22.58 | 30.87 |

Table 2: Model accuracy on selected quantum topics. Accuracy above 90% are shaded green and those below 50% are shaded red.

Models performing well on QC500 and QC1000 maintained comparable performance levels on larger benchmarks, suggesting that a carefully selected sample of a few hundred questions provides sufficient evaluation of quantum computing knowledge. Among the evaluated models, Claude 4 Sonnet achieved the highest overall performance, closely followed by GPT-5, GPT-4o, and Claude Sonnet 3.7. Notably, among open-source models, Phi-4-reasoning-plus, IBM Granite-3.3-8b-instruct, and Llama-3.1-8B-Instruct demonstrated reasonable performance on quantum computing tasks despite their smaller parameter counts. While these models still trail behind the larger proprietary systems, their relative competence suggests they could serve as practical starting points for domain-specific fine-tuning where computational resources are limited.

Table 2 shows a clear pattern: models handle basic concepts but decline sharply on advanced material, with the largest drop on quantum algorithms and security. Security questions were especially difficult, including recent work on phase mismatch attacks, crosstalk exploitation, QubitHammer, and quantum backdoors. These gaps highlight the challenge of fast moving areas that demand specialized knowledge, and the examples that follow illustrate the kinds of questions where even top models failed.

- What specific attack technique can manipulate the error rates of specific quantum gates ?

- What specific vulnerability does a quantum reorder attack exploit?

- What makes dynamical decoupling ineffective against QubitHammer attacks ?

This performance gap between foundational and advanced topics is particularly revealing. The disparity suggests that models have absorbed well-documented principles from extensive training data but struggle with recent developments where literature is sparser and terminology less standardized. The consistency of this pattern across model families, regardless of size or provider, indicates that the

challenge lies in the nature of the material rather than individual model limitations. Notably, even the expanded evaluation with agentic and deep research modes (Section 4.5) showed only marginal improvements on these advanced topics, confirming that web access alone cannot compensate for gaps in specialized reasoning. This finding has practical implications: users relying on LLMs for quantum computing assistance should exercise particular caution in rapidly evolving subfields where model knowledge may lag behind current research.

## 4.2 HUMAN PERFORMANCE BASELINE STUDY

To establish a human baseline for comparison with language model performance, we conducted a survey study with quantum computing researchers and practitioners. We carefully selected 30 questions from QC-Bench spanning different topic areas and complexity levels to assess human expertise across the quantum computing domain. The survey included questions from all seven categories. Participants were recruited from academic institutions and quantum computing research groups. Each respondent provided background information including their highest education level, years of experience in quantum computing, and age group. Further details on each participant's background and individual score are provided in the appendix, offering context for the distribution shown here. The sample questions below illustrate the style and difficulty of the survey items used in this comparison.

> **Sample Survey Questions**
>
> - Why is Shor's algorithm considered a threat to modern cryptographic security?
> - How does quantum transpilation optimize quantum circuits for real hardware?
> - Which quantum algorithm is specifically designed to process structured graph data?

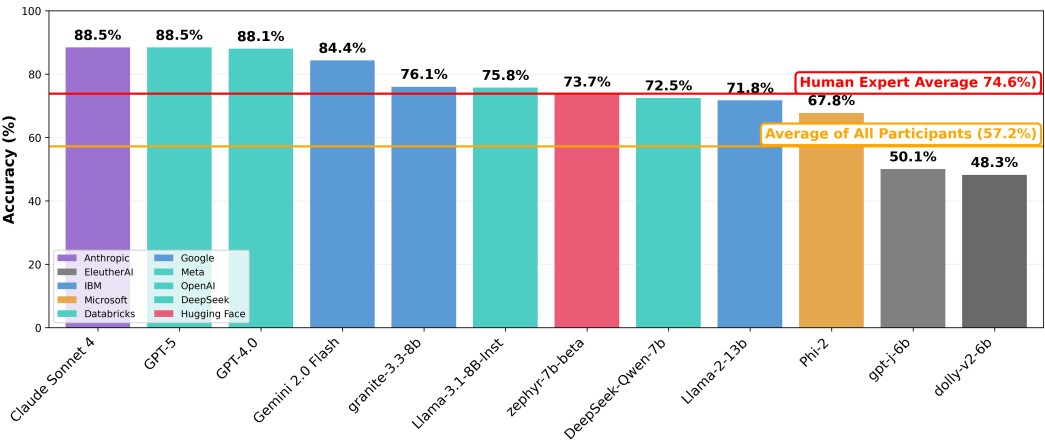

Figure 2: Performance comparison of selected LLMs across different capability tiers on the QC500 benchmark against human baselines. The visualization includes 12 representative models ranging from top performers to those scoring below novice human levels. Bars are colored by model provider.

Figure 2 presents a representative sample of LLM performance on the QC500 benchmark, showcasing models across the full performance spectrum. The majority of models shown exceed the all-participants average of 57.2%, while several surpass the expert average of 74.6%. The visualization highlights the dramatic performance range in quantum computing capabilities, from leading models like Claude 4 Sonnet (88.55%) and GPT-5 (88.07%) to models performing well below novice human levels, such as gpt-j-6b (50.14%) and dolly-v1-6b (48.29%). This selection demonstrates that quantum computing proficiency varies significantly across model families, sizes, and providers.

## 4.3 Performance Across Different Question Formats

Our evaluation extended beyond multiple-choice questions to assess model capabilities in diverse testing scenarios. For true/false questions, we modified the standard prompts to request binary verification of quantum computing statements. In open-ended questions we evaluated models' ability to generate explanations independently without options. Open-ended responses were graded based on factual correctness and conceptual completeness, with multiple-choice questions serving as the primary evaluation method while open-ended questions function as a supplementary diagnostic tool. Most models maintained strong performance on the true/false questions while showing clear degradation on open-ended assessments. Accuracy on the true/false set was tightly clustered, with smaller models often matching the large models once the task was reduced to a simple binary choice.

The limited number of options in the true/false evaluation leaves less room to distinguish stronger reasoning ability, so the gap between the very top systems and the weakest models nearly disappears in this format. By contrast, multiple-choice questions with four options revealed a more visible separation among high-end models, highlighting that true/false items are not an effective way to validate deeper research questions. Open-ended questions told a different story. GPT-5 not only produced the highest scoring answers when evaluated for correctness but also consistently provided richer, more contextually grounded explanations than its peers, and those detailed responses were closely aligned with the correct conclusions in most cases. This pattern underscores that open-ended evaluation exposes real differences in reasoning quality that are obscured when models face only binary decisions. Table 3 presents the complete results.

## 4.4 Fine-tuning Potential for Quantum Knowledge

We explored QC-Bench's utility for enhancing quantum computing capabilities through targeted fine-tuning. Using a subset of 4,400 questions for training and 1,000 questions as a test set, we fine-tuned five smaller language models using LoRA (Low-Rank Adaptation).

Our fine-tuning implementation used PyTorch with the Transformers library, applying LoRA with rank=8 and alpha=16 targeting attention projection matrices. We used a learning rate of 1e-4 with AdamW optimizer, batch size of 4 with gradient accumulation over 4 steps, and trained for a single epoch with warmup steps to ensure stable adaptation without overfitting.

Table 4 demonstrates the results across our selected models. Llama-3.1-8B-Instruct showed the strongest adaptation with a 5% improvement, while Gemma 2B achieved a modest 3.7% gain. Qwen1.5-MoE-A2.7B showed minimal improvement despite its Mixture-of-Experts architecture. Surprisingly, Phi-4-mini-reasoning experienced a slight performance decline, and EleutherAI/gpt-j-6b demonstrated a substantial 7% drop in accuracy. These mixed results highlight how model architecture significantly influences fine-tuning outcomes, with instruction-tuned models generally showing better adaptation to specialized quantum computing knowledge than their general-purpose counterparts.

| Model | T/F (%) | O-E (%) |
|---|---|---|
| GPT-5 | 93.27 | **89.07** |
| Claude Sonnet 4 | **93.99** | 88.84 |
| GPT-4o | 93.75 | 86.22 |
| Gemini 2.0 Flash | 92.31 | 84.09 |
| GPT-4.1 mini | 93.03 | 79.81 |
| llama-3.3-70b-versatile | 91.35 | 74.58 |
| Claude Haiku 3.5 | 93.75 | 78.15 |

Table 3: Accuracy on other question formats

| Model | Size | Before | After |
|---|---|---|---|
| Llama-3.1-8B-Instruc | 8B | 74.75 | 79.80 ↑ |
| Gemma 2B | 7B | 62.29 | 65.70 ↑ |
| Qwen1.5-MoE-A2.7B | 14.3B | 58.50 | 58.90 |
| Phi-4-mini-reasoning | 3.84B | 74.00 | 73.60 |
| EleutherAI/gpt-j-6b | 6B | 31.80 | 24.80 ↓ |

Table 4: Performance change after fine-tuning

Since effective fine-tuning remains feasible only for smaller models that consistently underperform relative to frontier systems, we shift focus to inference-time augmentation strategies. Retrieval-augmented generation and emerging agentic reasoning capabilities present promising alternatives, particularly for the largest models where parameter updates prove impractical. These approaches sidestep the weight-adaptation bottleneck by empowering models to dynamically query external sources and execute multi-step reasoning during inference, offering a more scalable pathway for specialized domain applications.

## 4.5 PERFORMANCE WITH AGENTIC AND DEEP RESEARCH MODES

Given the limited improvements observed from fine-tuning and the practical constraints of adapting larger models, we explored an alternative approach through agentic reasoning capabilities and deep research modes. These capabilities enable models to perform multi-step reasoning, search external sources, and synthesize information across multiple queries without requiring model adaptation or training data. This paradigm augments models with tool use and extended reasoning processes rather than modifying model weights to encode domain-specific knowledge. We evaluated frontier models equipped with their respective advanced capabilities: Claude Sonnet 4.5 Research Mode, Claude Sonnet 4.5 Extended Thinking, GPT-5.1 Deep Research, GPT-5.1 Agent Mode, and Gemini 3 Deep Research. These modes allow models to break down complex questions, search for relevant information, and synthesize answers through multi-step reasoning processes. We tested these capabilities on the QC500 subset, which provides a balanced evaluation across all quantum computing topics in our benchmark.

| Model | Before (%) | After (%) | Improvement |
|---|---|---|---|
| Claude Sonnet 4.5 (Research Mode) | 91.80 | 92.20 | +0.40 |
| Claude Sonnet 4.5 (Extended Thinking) | 91.80 | 92.60 | +0.80 |
| GPT-5.1 (Deep Research) | 91.40 | 92.80 | +1.40 |
| GPT-5.1 (Agent Mode) | 91.40 | 91.20 | -0.20 |
| Gemini 3 (Deep Research) | 87.40 | 89.20 | +1.80 |

Table 5: Performance of frontier models with advanced reasoning capabilities on QC500. Average improvement is 0.84 percentage points.

Results in Table 5 show the performance of these advanced reasoning modes. GPT-5.1 with Deep Research achieved the highest score at 92.80%, while both Claude Sonnet 4.5 variants performed above 92%. Gemini 3 Deep Research showed the largest improvement, gaining 1.8 percentage points to reach 89.20%. Notably, GPT-5.1 Agent Mode showed a slight decline of 0.2 percentage points compared to the base model. The average improvement across all models was 0.84 percentage points, demonstrating modest gains from these advanced capabilities.

These results indicate that while agentic and deep research modes provide measurable benefits, the improvements remain relatively modest on our benchmark. This suggests that the fundamental challenge in quantum computing knowledge assessment lies in the breadth and depth of knowledge encoded during pretraining rather than in reasoning capabilities alone. Questions in QC-Bench are designed to test factual knowledge and conceptual understanding rather than multi-step reasoning, which may explain why reasoning-augmented modes show limited advantage. However, advanced reasoning modes offer key advantages: (1) they require no training data or computational resources for model adaptation, (2) they can access current information beyond the model's knowledge cutoff, and (3) they can be applied to the largest and most capable models where fine-tuning is often impractical.

## 4.6 MULTILINGUAL BENCHMARK PERFORMANCE

To investigate how quantum computing knowledge transfers across languages, we evaluated all models on Spanish and French translations of QC500. This experiment provides quantitative insights into linguistic generalization of specialized technical knowledge. Figure 3 shows Spanish versus French accuracy for selected models. While most models fall along a diagonal cluster indicating correlated cross-lingual performance, the distribution reveals systematic language-dependent performance gaps. Across our full benchmark set, models lose on average 11.2 percentage points in French and 15.2 percentage points in Spanish relative to English baselines. This asymmetry is particularly notable, as Spanish exhibits approximately 55 percent greater performance degradation than French. Remarkably, only 34.5% of models maintain scores above 75% in Spanish, compared to 44.8% in French and 69.0% in English. The most linguistically consistent models (Claude 4 Sonnet, GPT-5, and Gemini 2.0 Flash) show standard deviations below 0.6 across languages, while the least consistent (Phi-4-reasoning) exhibits a standard deviation of 31.2.

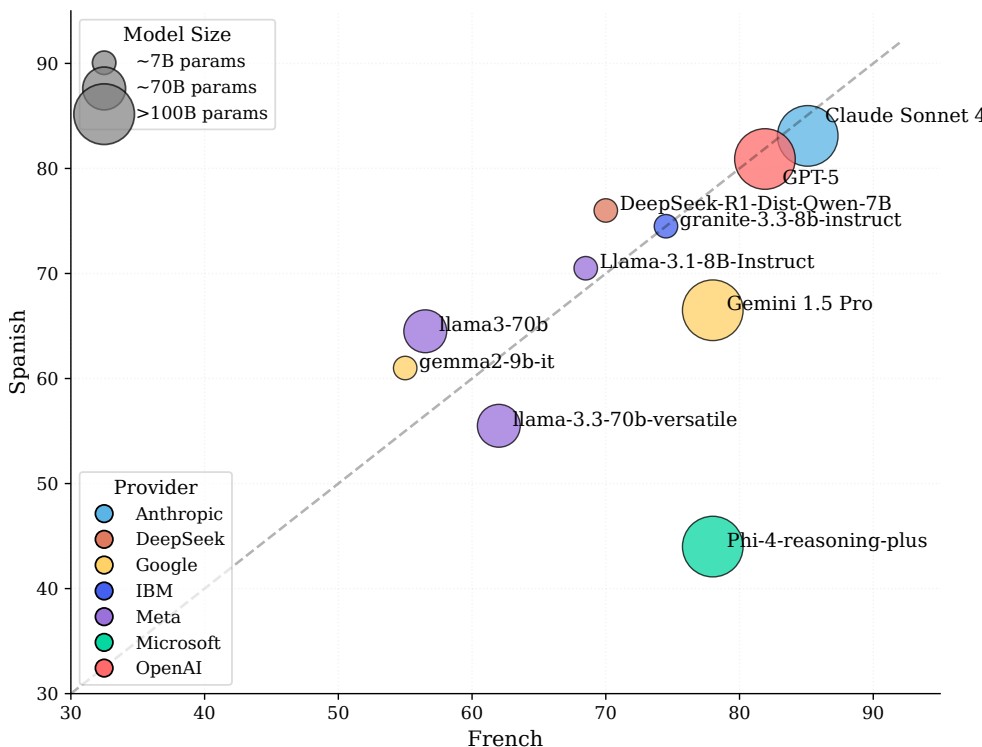

Figure 3: Bubble chart of Spanish (vertical) versus French (horizontal) accuracy on the QC500 benchmark. Each bubble's area is proportional to the model's parameter count; colors indicate providers. The diagonal dashed line marks equal performance across the two languages. Bubbles below the line signal larger accuracy loss in Spanish.

## 5 DISCUSSION

Our evaluation reveals a clear performance pattern across all tested models: strong results on foundational topics with significant decline on advanced domains. Top models achieve over 92% accuracy on basic quantum concepts but drop below 77% for quantum security questions. This performance drop is particularly evident in questions about emerging attack vectors like phase mismatch attacks and QubitHammer, where even the most advanced models failed to provide accurate responses consistently. Notably, leading LLMs outperform many practitioners and experts in our human survey, where performance ranged from 26.6% to 86% depending on education level and experience (detailed results in the appendix). In addition, the results highlight a widening gap between recent state-of-the-art LLMs and smaller models, a trend that persists even after fine-tuning. These high-capacity systems show clear advantages not only on complex multiple-choice tasks but especially on open-ended questions, where they deliver more accurate and detailed explanations. Smaller models, by contrast, plateau despite fine-tuning, indicating that model scale and training pipelines remain critical for strong performance on demanding quantum computing assessments.

Question format comparison shows GPT-5 maintaining 89.07% accuracy on open-ended quantum explanations while most competitors show degradation without multiple-choice options. This suggests many models rely on recognizing answer patterns rather than constructing explanations from fundamental understanding. Our multilingual testing reveals concerning disparities, with average performance dropping 11.2 percentage points in French and 15.2 points in Spanish. Fine-tuning results demonstrate significant variation in how models adapt to quantum knowledge. Llama-3.1-8B-Instruct improved by 5.3% through fine-tuning, while EleutherAI/gpt-j-6b declined by 7%, suggesting that instruction-tuned models more readily incorporate specialized quantum knowledge.

As quantum computing advances toward practical implementation, retrieval-augmented generation can complement fine-tuning, particularly since practical fine-tuning is mainly feasible for smaller models. While targeted fine-tuning can modestly improve accuracy for compact systems, it remains costly and inflexible for the larger architectures that already set the performance frontier. Retrieval-augmented generation, by contrast, allows those high-capacity LLMs to access curated domain sources and continuously updated technical literature, avoiding the need for repeated full retraining.

## 6 LIMITATIONS AND FUTURE WORK

QC-Bench offers a comprehensive evaluation of quantum computing knowledge, with English as the primary language and a large QC500 subset already available in Spanish and French. A next step is to expand coverage beyond QC500 by translating a larger portion of the benchmark into Spanish and French, and by adding more languages to better reflect global practice. Additional work includes increasing the diversity of non-English source materials and assessing cross-lingual consistency to provide a more complete view of multilingual performance.

Our evaluation relies primarily on accuracy as the central performance metric, which effectively captures models' factual knowledge but may not fully represent their conceptual understanding or reasoning capabilities. We chose accuracy for its interpretability, directness, and alignment with our goal of measuring factual correctness in quantum computing knowledge. Future research could explore alternative metrics such as calibration scores for confidence assessment, partial credit scoring for near-correct responses, or semantic similarity measures for evaluating open-ended explanations beyond binary correctness judgments.

Additionally, incorporating statistical frameworks such as error bars and confidence intervals would enhance the interpretability of results, as discussed by Miller (2024). Given the extensive nature of our evaluation across 31 models, multiple question formats, three languages, fine-tuning experiments, and agentic evaluation modes, incorporating this level of statistical rigor was beyond the current scope. We leave this as a direction for future work.

## 7 CONCLUSION

As Large Language Models (LLMs) are increasingly tasked with reading, explaining, and answering questions about quantum computing literature, rigorous domain evaluation is essential. QC-Bench provides a comprehensive assessment with 5,400 multiple-choice items plus 416 true/false and 421 open-ended questions across seven core domains. Across 31 systems, we find a consistent pattern: strong results on foundational material but marked drops on advanced topics. Top systems clear 92% on basic concepts yet fall below 77% on security questions, including items on recent developments in quantum security. Format matters: many models score well on multiple choice but degrade on open-ended responses; GPT-5 maintains the strongest open-ended performance among evaluated systems (89.07%) and produces more detailed, context-grounded explanations. Relative to human baselines (23.3%–86.7%), eight models exceed the all-participants average of 74.6% and surpass the expert average of 80.0%. Multilingual testing shows asymmetry, with average accuracy declines of 11.2 points in French and 16.2 in Spanish relative to English, indicating that quantum knowledge does not transfer uniformly across languages.

Methodologically, the results indicate a widening gap between state-of-the-art, high-capacity systems and smaller models, a difference that persists even after fine-tuning. Gains from fine-tuning are modest, typically only a few percentage points, and can sometimes reduce accuracy, making the computational cost difficult to justify for larger architectures. Evaluation of frontier models with agentic and deep research capabilities showed an average improvement of only 0.84 percentage points even with full internet access, suggesting that web search alone cannot compensate for gaps in specialized technical reasoning. Alternative approaches for enhancing performance on demanding technical domains remain an open research question. Quantum computing remains one of the most demanding areas for language models, and continued evaluation of LLM capabilities in this domain is essential for tracking progress and ensuring reliable performance as the field evolves.

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

## A  APPENDIX

### A.1  QUESTION EXTRACTION AND GENERATION

For the QC500 and QC1000 subsets, researchers systematically reviewed the selected quantum computing papers and identified key concepts, algorithms, and principles that capture both foundational and advanced material. Rather than copying sentences directly, each question was crafted by rephrasing important findings and definitions from the literature to create original items while preserving scientific accuracy. Draft questions underwent multiple rounds of verification for technical correctness and clarity to ensure they tested understanding rather than memorization of phrasing. In parallel, we leveraged language models in a controlled setting to suggest candidate questions from the same source papers, but every suggestion was filtered and rewritten by researchers to maintain consistency with the human-validated style and difficulty. This combined process produced a balanced set of high-quality questions that reflect authentic quantum computing research while supporting rigorous evaluation across diverse topics and difficulty levels. The question creation process involved developing multiple answer options for each extracted concept. Below we illustrate this process with an example:

---

**Example: Question Development Process**

**Initial concept from paper:** Quantum circuit synthesis involves decomposing unitary operations into implementable gate sequences.

**Generated question:** What is the primary purpose of quantum circuit synthesis?

**Initial answer options (6 generated):**

  A. To convert a quantum circuit into a classical circuit by removing superposition properties

  B. To merge multiple unitary matrices into a single high-dimensional operator without gate decomposition

  C. To decompose a unitary matrix representing the circuit into a sequence of gates from the native gate set

  D. To encode classical information into qubit states without performing any gate-level modifications

  E. To simulate quantum circuits on classical computers using tensor networks

  F. To optimize quantum algorithms for specific hardware architectures

**Final selection (4 options):** Options E and F were eliminated as they describe related but distinct processes. The final question includes the correct answer (C) and three plausible distractors that test understanding of quantum circuit concepts.

---

**Quantum Circuit Synthesis**

**Question:** What is the primary purpose of quantum circuit synthesis?

  A. To convert a quantum circuit into a classical circuit by removing superposition properties

  B. To merge multiple unitary matrices into a single high-dimensional operator without gate decomposition

  C. To decompose a unitary matrix representing the circuit into a sequence of gates from the native gate set

  D. To encode classical information into qubit states without performing any gate-level modifications

**Answer:** C

## A.2 AUTOMATED QUESTION MINING FROM RESEARCH PAPERS

To expand our benchmark beyond human-authored questions, we employed Large Language Models (LLMs), specifically Gemini 2.0 Flash, Gemini 1.5 Pro, GPT-4.0, and Claude 3.7 Sonnet, to extract additional questions from 212 carefully selected quantum computing papers. This automated extraction process generated an initial pool of over 8,000 candidate questions, each with six potential answer options.

The filtering process involved multiple stages. First, we identified and removed questions that strayed from quantum computing into adjacent domains. Examples of filtered questions include:

---

**Example: Filtered Question - General Cybersecurity**

**Question:** Which encryption protocol is most commonly used for securing HTTP connections?

    A.  TLS/SSL

    B.  SSH

    C.  IPSec

    D.  WPA2

*Reason for filtering:* While encryption is relevant to quantum cryptography, this question addresses classical network security without quantum computing connection.

---

**Example: Filtered Question – Mathematical Modeling**

**Question:** In the analysis of an ordinary differential equation system, what does a non-positive log-norm of the coefficient matrix imply?

    A.  The system is unstable for all inputs

    B.  The matrix has only imaginary eigenvalues

    C.  The solution decays or remains bounded over time

    D.  The matrix is diagonalizable over the complex field

*Reason for filtering:* While this concept appears in resource analyses for quantum-inspired algorithms, it tests classical stability theory in differential equations and does not assess quantum computing knowledge.

---

After removing duplicate questions, filtering irrelevant content, and conducting manual quality review, we retained 4,400 high-quality questions. For each retained question, we selected the four most relevant answer options from the initial six, ensuring each question had one correct answer and three well-crafted distractors that effectively test quantum computing knowledge.

## A.3 QUESTION TRANSLATION AND MULTILINGUAL VALIDATION

For the QC500 subset, we created Spanish and French translations using a multi-stage process. We employed the same four LLMs (Gemini 2.0 Flash, Gemini 1.5 Pro, GPT-4.0, and Claude 3.7 Sonnet) to generate initial translations. A typical translation prompt was structured as follows:

```
"Translate the following quantum computing question from English to French,
maintaining technical accuracy and appropriate scientific terminology:
[Question and answer options]"
```

For each question, we collected translations from all four models and selected the most accurate version. This selection was then reviewed by individuals proficient in both languages who verified technical terminology and ensured conceptual accuracy. The translation process preserved the semantic content while adapting to language-specific conventions for scientific terminology.

Examples of translated questions include:

---

**French Translation Example 1**

**Question:** Pourquoi les attaques par impulsion à grande échelle sont-elles difficiles à réaliser dans les systèmes partagés ?

- A. Elles dépendent d'un accès chiffré aux qubits
- B. Elles nécessitent un accès à la machine au niveau administrateur
- C. Elles requièrent de nombreux qubits, auxquels les utilisateurs n'ont généralement pas accès
- D. Elles échouent si la machine est calibrée

**Answer:** C

---

**French Translation Example 2**

**Question:** Pourquoi les algorithmes quantiques paramétriques sont-ils difficiles à vérifier sémantiquement ?

- A. Ils utilisent des paramètres fixes définis dans le matériel
- B. Ils reposent uniquement sur un post-traitement classique
- C. Leurs paramètres entraînés manquent d'interprétabilité inhérente
- D. Leur structure est identique pour tous les ensembles de données

**Answer:** C

---

**Spanish Translation Example 1**

**Question:** ¿Cuál es la principal diferencia entre la privacidad diferencial clásica y la privacidad diferencial cuántica?

- A. La PD cuántica extiende las garantías de privacidad a estados cuánticos indistinguibles utilizando distancias de traza
- B. La PD cuántica elimina la necesidad de análisis probabilístico
- C. La PD cuántica se aplica solo a registros de qubits entrelazados
- D. La PD cuántica se impone eliminando los resultados de medición de qubits

**Answer:** A

---

**Spanish Translation Example 2**

**Question:** ¿Qué algoritmo clásico se utiliza comúnmente después del paso cuántico del Algoritmo de Shor?

- A. Algoritmo de Dijkstra
- B. Expansión de fracciones continuas
- C. Integración de Monte Carlo
- D. Búsqueda binaria

**Answer:** B

---

### A.4 QUESTION FORMAT DIVERSIFICATION

To evaluate models' performance across different cognitive tasks, we expanded our benchmark with true/false and open-ended questions. For this expansion, we selected 40 additional research papers to ensure diverse content and avoid repetition. Both LLMs and human experts generated questions following similar protocols to the initial question creation phase.

This process yielded 416 true/false questions and 421 open-ended questions. True/false questions were created by converting factual statements into binary assessments:

---

**True/False Question Examples**

**Question:** The Bloch sphere is a geometrical representation of pure quantum states of a two-level quantum mechanical system.
**Answer:** True

**Question:** Dirac notation can only represent pure states, not mixed states.
**Answer:** False

**Question:** The E91 protocol is based on entangled particles and provides a method for secure quantum key distribution.
**Answer:** True

**Question:** Quantum error correction codes do not require any additional qubits beyond the physical qubits used to represent the logical qubit.
**Answer:** False

---

Open-ended questions were designed to assess deeper understanding and explanatory capabilities:

---

**Open-Ended Question Examples**

**Question:** What is the no-cloning theorem and its implication for quantum information?

**Question:** How does the Heisenberg uncertainty principle affect the measurement of quantum states?

**Question:** In the context of quantum states, what distinguishes a pure state from a mixed state?

**Question:** Explain the significance of the CNOT gate in quantum entanglement.
**Sample Answer:** The CNOT gate, or controlled-NOT gate, is crucial for creating entanglement between two qubits, as it flips the state of the target qubit only if the control qubit is in the state $|1\rangle$.

---

For each open-ended question, we developed sample answers to facilitate consistent evaluation across different models. These questions were assessed manually to determine whether model responses captured the essential concepts and technical accuracy required for each topic.

### A.5 HUMAN PERFORMANCE BASELINE STUDY

Table 6 reports accuracies for the first 20 respondents. Scores range from 23.3% to 86.7%, with an overall average of 57.2%. Participants with 5+ years of experience achieved an average of 79.4%, providing a reference point for expert-level performance. Education level shows a clear pattern: all PhD-trained participants scored at or above 73.3%, while no BS-level participant reached 60%. These results provide a concrete reference distribution for interpreting model-human comparisons in the main results.

### REFERENCES FOR EACH TOPIC

Table 7 lists the literature sources and citation coverage for all seven benchmark topics. Rapidly developing areas such as quantum cybersecurity and quantum machine learning rely heavily on the most recent papers to capture ongoing advances, whereas foundational categories such as quantum theory and quantum error correction draw on a broader historical record to reflect the principles that remain central to the discipline. This distribution ensures that the benchmark balances up-to-date research with enduring theoretical foundations, giving a clear view of how source material supports each topic area.

| Participant | Education | Experience | Age Group | Score | Accuracy |
|---|---|---|---|---|---|
| P1 | MS | 2–5 yrs | 25–35 | 19/30 | 63.3% |
| P2 | BS | <1 yr | 18–25 | 14/30 | 46.7% |
| P3 | PhD | 5+ yrs | 35–45 | 25/30 | 83.3% |
| P4 | MS | 1–2 yrs | 25–35 | 21/30 | 70.0% |
| P5 | PhD | 2–5 yrs | 35–45 | 23/30 | 76.7% |
| P6 | BS | <1 yr | 18–25 | 12/30 | 40.0% |
| P7 | MS | 1–2 yrs | 25–35 | 17/30 | 56.7% |
| P8 | MS | 5+ yrs | 35–45 | 24/30 | 80.0% |
| P9 | PhD | 2–5 yrs | 25–35 | 22/30 | 73.3% |
| P10 | BS | 1–2 yrs | 18–25 | 16/30 | 53.3% |
| P11 | PhD | 5+ yrs | 45–55 | 26/30 | 86.7% |
| P12 | BS | <1 yr | 18–25 | 11/30 | 36.7% |
| P13 | MS | 2–5 yrs | 25–35 | 20/30 | 66.7% |
| P14 | PhD | 1–2 yrs | 25–35 | 22/30 | 73.3% |
| P15 | BS | <1 yr | 18–25 | 8/30 | 26.7% |
| P16 | PhD | 5+ yrs | 35–45 | 25/30 | 83.3% |
| P17 | MS | 2–5 yrs | 25–35 | 23/30 | 76.7% |
| P18 | PhD | 5+ yrs | 45–55 | 24/30 | 80.0% |
| P19 | BS | <1 yr | 18–25 | 7/30 | 23.3% |
| P20 | MS | 1–2 yrs | 25–35 | 18/30 | 60.0% |
| **Expert Average (5+ years experience):** | | | | | **79.4%** |
| **All Participants Average:** | | | | | **57.2%** |

Table 6: Human participant survey results on 30-question quantum computing assessment

## A.6 FINE-TUNING METHODOLOGY

Our fine-tuning experiments employed Low-Rank Adaptation (LoRA) to efficiently adapt smaller language models to quantum computing knowledge while maintaining computational feasibility. The implementation utilized a carefully selected subset of 4,167 question-answer pairs from the QC-Bench dataset for training, with an additional 1,000 questions reserved for evaluation. The training data was formatted as concatenated prompt-completion pairs to maximize learning efficiency within context length constraints. We applied LoRA with rank 8 and alpha 16, specifically targeting the attention projection matrices (q_proj, k_proj, v_proj, o_proj) which are critical for knowledge representation. The training configuration employed a batch size of 4 with gradient accumulation over 4 steps, resulting in an effective batch size of 16, paired with a conservative learning rate of 1e-4 using the AdamW optimizer. To ensure stable convergence, we implemented 50 warmup steps followed by training for a single epoch, which empirical testing showed was sufficient to achieve knowledge transfer without overfitting. The models were loaded in FP16 precision to reduce memory requirements while maintaining numerical stability, with automatic device mapping to optimize GPU utilization. Early stopping was monitored through validation accuracy computed every 200 steps, though most models converged within the single epoch. This approach resulted in training only approximately 0.5-2% of total model parameters, demonstrating that quantum computing knowledge can be effectively incorporated through targeted parameter updates rather than full model retraining.

| Topic | References | Years | # |
|---|---|---|---|
| Basic Concepts | Aharonov et al. (1998); Terashima & Ueda (2005); Arrazola et al. (2022); Williams & Gray (1998); Hayward (2008); Xu et al. (2023); Peterer et al. (2015); Schuld & Killoran (2019); Gudder (1983); Biard et al. (2021); Kowalski & Bauman (2023); Del Santo & Gisin (2025); Younis & Iancu (2022); Hua et al. (2023); Sikorski (2023); Nielsen & Chuang (2010); Preskill (2018); Deutsch (1985); Feynman (1982); Kjaergaard et al. (2020); Bharti et al. (2022); Zurek (2003); Deutsch & Ekert (1998); Bennett et al. (1993); Quinton et al. (2025); Phillipson (2024); Cirac & Zoller (1995); Benioff (1980); Giovannetti et al. (2008); AbuGhanem (2025); Farhi et al. (2000); DiVincenzo (2000); Lloyd (1996); Knill et al. (2001); King et al. (2025); Halimeh et al. (2025); Puig et al. (2025); Munro et al. (2005); Harrow & Leung (2004); Steane (1996) | 1980–2025 | 39 |
| Gates & Circuit Design | Zhang et al. (2022b); Ren et al. (2024); Peham et al. (2022); Kusyk et al. (2021); Ostaszewski et al. (2021); Rosa et al. (2025); DiVincenzo (1998); Kalloor et al. (2024); Senapati et al. (2024); Cao et al. (2012); Venturelli et al. (2018); Barenco et al. (1995); Vandersypen et al. (2001); Steane (1999); Laflamme et al. (2002); Cory et al. (2000); Cross et al. (2019); Linke et al. (2017); Smith et al. (2019); Maslov et al. (2008); McKay et al. (2018); Chong et al. (2017); Hashim et al. (2021); Zulehner et al. (2018); Wille et al. (2019); Murali et al. (2019) | 1995–2025 | 26 |
| Quantum Machine Learning | Wittek (2014); Bowles et al. (2024); Vishwakarma et al. (2024); Ranga et al. (2024); Rath & Date (2024); Biswas (2025); Bischof et al. (2025); Kreplin & Roth (2024); Chinzei et al. (2024); Afane et al. (2025); Yu et al. (2024); Schuld et al. (2015); Havlíček et al. (2019); Cerezo et al. (2021); Biamonte et al. (2017); Schuld & Petruccione (2018); Farhi & Neven (2018); Dunjko & Briegel (2018); Benedetti et al. (2019); Lloyd & Weedbrook (2018); Beer et al. (2020); Huang et al. (2021); Mitarai et al. (2018); Rebentrost et al. (2014); Grant et al. (2018); Cong et al. (2019); Schuld et al. (2020); Amin et al. (2018); Perdomo-Ortiz et al. (2018); Moll et al. (2018); Arrazola et al. (2020); Romero et al. (2017); Tacchino et al. (2019); Li et al. (2018); Abbas et al. (2021) | 2014–2025 | 35 |
| Distributed Computing | Cuomo et al. (2020); Cacciapuoti et al. (2020); Wehner et al. (2018); Kimble (2008); Simon (2017); D'Adamo et al. (2022); Dahlberg et al. (2019); Caleffi & Cacciapuoti (2020); Pompili et al. (2021); Simon (2017); Van Meter (2016); Munro et al. (2015); Meter et al. (2013); Van Meter et al. (2009); Lloyd (1993); Cirac et al. (1997); Perseguers (2013); Pant et al. (2019); Ishizaka & Hiroshima (2008); Simon (2015); Laurat et al. (2005); Avis et al. (2019); Van Meter et al. (2020); Joshi et al. (2020); Lemos et al. (2014); Pirandola et al. (2018); Azuma et al. (2022); Takeda & Furusawa (2023); Joshi et al. (2024); Khatri & Wilde (2021); Bhaskar et al. (2020); Askaridis et al. (2021); Chi et al. (2022); Kozlowski et al. (2020); Muralidharan et al. (2016) | 1993–2024 | 35 |
| Quantum Security | Dhar et al. (2024); Mehic et al. (2023); Chu et al. (2023); Zhao et al. (2024); Xu et al. (2023); Krawec et al. (2024); Zhang et al. (2022a); Xu & Szefer (2024); Tan et al. (2025); Sahu & Mazumdar (2024); Ralegankar et al. (2021); Kalaivani et al. (2021); Pirandola et al. (2020); Bernstein & Lange (2017); Lo et al. (2014); Bennett & Brassard (2014); Xu et al. (2020); Ekert (1991); Bennett (1992); Scarani et al. (2009); Lo & Chau (1999); Shor & Preskill (2000); Mayers (2001); Renner (2008); Wootters & Zurek (1982); Diamanti et al. (2016) | 1982–2025 | 26 |
| Error Correction | Fowler et al. (2012); Shor (1995); Lidar & Brun (2013); Terhal (2015); Aharonov & Ben-Or (2008); Chiaverini et al. (2004); Reed et al. (2012); Bombín & Martin-Delgado (2006); Gottesman (1997); Nielsen & Flensberg (2021); Steane (1996); Kitaev (1997); Preskill (1998); Bacon (2006); Aliferis et al. (2006); Calderbank & Shor (1996); Steane (1999); Dennis et al. (2002); Acharya et al. (2024); Barends et al. (2014); Kelly et al. (2015); Cory et al. (1998); DiVincenzo & Shor (1996); Bravyi & Kitaev (2005); Albert et al. (2018); Bennett et al. (1996); Gambetta et al. (2017); McEwen et al. (2023); Takita et al. (2017) | 1995–2024 | 29 |
| Quantum Algorithms | Montanaro (2016); Mosca (2008); Childs & Van Dam (2010); Hastings et al. (2014); Gheorghiu & Mosca (2025); Krovi (2023); Jin et al. (2023); Qiang et al. (2021); Benedetti et al. (2021); Du et al. (2022); Motta et al. (2020); Grover (1996); Shor (1999); Harrow et al. (2009); Ambainis (2007); Kitaev (1995); Nayak & Wu (1999); Childs et al. (2003); Cleve et al. (1998); Farhi et al. (2000); Nielsen & Chuang (1997); Aharonov et al. (2008); Bennett et al. (1997); Deutsch & Jozsa (1992); Nielsen & Chuang (2002); Brassard et al. (1997); Jordan (2005); Reichardt (2009); Wiebe et al. (2012); Aaronson & Arkhipov (2011) | 1995–2025 | 30 |

Table 7: Topic Coverage and Source Papers