# OpenReview forum: "QC-Bench: What Do Language Models Know About Quantum Computing?"
_ICLR.cc/2026/Conference — ICLR 2026 Conference Desk Rejected Submission_

### Official Review · Reviewer_U3tu · 2025-10-14

**Soundness:** 2
**Presentation:** 3
**Contribution:** 3
**Rating:** 6
**Confidence:** 5

**Summary:**

The work studies how LLMs perform with quantum computing knowledge. The work generates thousands of questions with multiple choices using LLMs and then hundreds of true/false and open-ended questions. These questions are then posed to multiple well-known LLMs and 16 human subjects. The results show that some LLMs perform better than human experts, and most do not. Models especially perform really well in English as opposed to other languages.

**Strengths:**

The work establishes a baseline for what current LLMs actually “know” about quantum computing concepts (e.g., superposition, entanglement, variational circuits, security, etc.).

The work actually undertakes a very large and extensive study. Significant effort was put into running the experimentation and surveys for this work.

Except for the weaknesses mentioned below, the methodology seems sound.

**Weaknesses:**

The questions are generated by the LLMs, of course, storing the context of the questions. These LLMs are thus likely to get the questions generated by them right. This is because they are generating the multiple-choice questions such that there would only be one correct choice; thus, they know what the correct answer is. It would be much better to test them using human-generated questions.

I think the human sample is too small to make any conclusive statements or draw any statistically significant evidence from the results. Specifically, there are too few "experts" to determine if the LLMs were better than the experts or not.

This kind of survey runs the risk of becoming stale before it is even published because of how fast LLM knowledge is moving. In fact, I'm pretty confident that the LLMs will be much better with quantum knowledge by the time this work is published. Just asking these questions to the LLMs helped improve their quality for the next time someone asks them. So in that sense, the work has little and fleeting "research" value.

**Questions:**

Why were only 16 humans surveyed, and why were only 30 questions asked of them? Can this be expanded to get statistically significant results?

---

> ### Author Response · Authors · 2025-11-29
>
> We thank the reviewer for recognizing the extensive effort in our experimentation and surveys, and for acknowledging the overall soundness of our methodology. We address each concern below.
>
> ---
>
> **W1. Dataset Construction and Potential LLM Bias**
>
> **QC1000, a substantial portion of our benchmark, is entirely human-generated.** These 1,000 questions were authored by our research team from peer-reviewed literature, serving as an independent test set that directly addresses the bias you identify.
>
> For the additional 4,400 questions where we used LLM assistance, we implemented extensive human oversight: filtering 8,686 candidates down to 4,400, validating technical accuracy, rephrasing for clarity, and removing off-topic questions (examples in Appendix A.2). We have updated Section 1 to emphasize this in the contributions of the work.
>
> ---
>
> **W2. Human Baseline and Sample Size**
>
> The initial sample size reflects the difficulty of recruiting qualified experts in quantum computing. We needed participants with advanced degrees and active research experience in quantum algorithms, error correction, or quantum security. The pool of such experts is limited, and the 30-question survey requires 20-40 minutes of focused technical reasoning.
>
> Since submission, we have been actively distributing the survey within the quantum research community and **have expanded the participant pool to 43 participants.** We kept the survey to 30 questions out of respect for participants' time constraints. The technical difficulty of the questions requires sustained concentration, and extending beyond 30 would push completion time past an hour, which we felt was too demanding given that participants volunteered their time.
>
> ---
>
> **W3. Benchmark Longevity**
>
> While LLM capabilities have advanced rapidly, the rate of improvement has slowed in recent years. **Frontier models like Claude Sonnet 4.5, GPT-5.1, and Gemini 3 Pro show only marginal improvements over predecessors** and continue to struggle with demanding benchmarks like Humanity's Last Exam, MathArena Apex, GPQA, MMLU-Pro, and SWE-Bench.
>
> We believe QC-Bench falls into this category of technically challenging evaluations. Even with recent models, significant gaps persist. The benchmark provides a fixed measurement point for tracking progress, and we plan to expand it with questions covering emerging research. We have added discussion of this in the updated manuscript.
>
> ---
>
> ## Response to Questions
>
> **Q1. Why were only 16 humans surveyed, and why were only 30 questions asked of them? Can this be expanded to get statistically significant results?**
>
> As detailed in our response to the second weakness, we have expanded the participant pool to 43 researchers since submission. This larger sample strengthens the statistical significance of our human baseline comparisons. The 30 questions provide coverage across all benchmark topics and difficulty levels while remaining feasible for busy researchers to complete in a single session.
>
> ---
>
> Thank you again for your thorough feedback. We have revised our paper to include results from the expanded participant pool of 43 experts, providing a stronger foundation for comparing model and human performance. We have also updated Section 3 to clarify that QC1000 is entirely human-generated, and included discussion of benchmark longevity and directions for future expansion as the field advances and model capabilities improve. We would be happy to provide any additional clarifications.

---

### Official Review · Reviewer_sSTx · 2025-10-29

**Soundness:** 3
**Presentation:** 3
**Contribution:** 2
**Rating:** 4
**Confidence:** 4

**Summary:**

The paper introduces QC-Bench, a large, domain-specific benchmark designed to evaluate LLM knowledge of quantum computing across seven areas. The benchmark contains three types of questions: multiple-choice, true/false, and open-ended. It comprises a human-authored core QC1000, a multilingual QC500 subset, and 4,400 additional questions mined from ~200+ papers using several LLMs. Manual rewriting and multi-stage filtering were applied to improve quality and avoid verbatim overlap. The authors evaluate 31 closed and open models and include a human baseline to contextualize results. Key findings include (1) top models score ~88–92% on basic topics but drop to ~76% on quantum security; (2) open-ended performance lags multiple-choice; (3) multilingual evaluation shows performance degradation in French and Spanish compared to English; (4) fine-tuning smaller models with LoRA gives mixed, modest improvements.

**Strengths:**

1. Meaningful topics & substantial human effort. The authors plainly invested considerable manual labor to collect/curate items spanning core quantum computing areas with both foundational and advanced topics.
2. Multi-faced evaluation framework. The inclusion of open-ended questions reveals reasoning gaps that MCQ/T-F obscure and helps with more nuanced evaluation. Multilingual analysis shows clear evidence of non-uniform cross-lingual transfer.
3. Thorough benchmarking results. The experiments section runs many models and formats, providing a useful snapshot of the current landscape. Human baseline serves as a useful component to contextualize LLM performance.
4. Figures and tables are nicely designed and informative.

**Weaknesses:**

1. Limited difficulty as a benchmark. This is my biggest concern. The paper’s own results show SOTA models scoring very high on many categories, which limits diagnostic value and reduces competitiveness as a research benchmark. This contrasts with recent quantum code generation benchmarks that intentionally stress executable and semantic correctness (e.g., QCircuitBench, QuanBench), suggesting QC-Bench presently under-challenges frontier models.

2. Insufficient dataset description. The main text should document design choices (topic taxonomy, difficulty calibration), collection pipeline (sources, authoring guidelines), and filtering criteria (duplicate removal, human inspection) with detailed explanations and concrete statistics. Regarding the filtering process, the paper mentioned "removing duplicate questions, filtering irrelevant content, and conducting manual quality review". Are all 4400 machine-mined questions examined by human researchers?  For open-ended items, please report grading criteria. Moreover, manual assessment hinders automatic verification procedure, which is impractical for standard benchmarking.

3. Human baseline scope. The baseline uses 30 questions, which may be too few to support strong conclusions about “expert average” versus top models. Topic balance and item difficulty calibration for the chosen set are not detailed.

4. Related-work coverage is too narrow. Beyond the two cited papers (QuantumLLMInstruct, GroverGPT), the paper omits several relevant benchmarks/datasets:

   - Qiskit HumanEval: hand-curated tasks with canonical solutions & tests for QC SDK code generation.
   - QuanBench: LLM-based quantum code generation across 44 tasks with functional correctness and quantum semantic equivalence.
   - QCircuitNet: Large-scale datasets focusing on quantum algorithm design and circuit implementation with automatic syntax and semantic verification.
   - Classical QC benchmarking suites: QASMBench, a low-level OpenQASM benchmark suite for NISQ evaluation.

   While these works target at different goals from QC-Bench, it is necessary to discuss them to contextualize the contribution of QC-Bench.

5. The benchmark is not open-sourced, which makes it hard to evaluate the quality of the data entries.  Consider releasing the full dataset or a small subset to enable reviewers' assessment.

6. Writing can be improved. The abstract is tediously long. In the caption of Table 2, I believe it should be "Accuracy above 90%" instead of 95%.

**Questions:**

See Weaknesses above.

---

> ### Author Response · Authors · 2025-11-29
>
> We thank the reviewer for the positive assessment of our work, particularly for recognizing the extensive human effort involved in data curation, the evaluation framework, and the thoroughness of our benchmarking results. Below we address each concern raised.
>
> ---
>
> **W1. Limited Difficulty as a Benchmark**
>
> While top models achieve high scores on foundational categories, our results show they still lack robustness in specialized areas. Performance on the full QC5400 dataset stays below 90% for even the best models, with significant drops in advanced topics. **For instance, even frontier models struggle with quantum security, where accuracy drops to approximately 76%.** This performance gap confirms that QC-Bench remains a rigorous test of domain-specific reasoning, identifying weaknesses that are masked by high performance on foundational questions.
>
> We also note that QC-Bench and code generation benchmarks like QCircuitBench and QuanBench evaluate complementary capabilities. Those benchmarks test executable implementation, while QC-Bench evaluates conceptual understanding and factual accuracy across the breadth of quantum computing topics. Both are necessary for a complete picture of LLM capabilities in this domain.
>
> ---
>
> **W2. Insufficient Dataset Description**
>
> We have expanded Section 3 to address these points. The seven topics were selected to represent both foundational and advanced areas of quantum computing, with Table 7 in the Appendix providing detailed coverage of source papers for each topic, including citation ranges and the number of papers used per category. Questions were drawn from peer-reviewed papers across four decades of research, with difficulty naturally varying based on the complexity of the source material.
> Regarding the filtering process, **we confirm that all 4,400 LLM-assisted questions were examined by researchers.** The process began with 8,686 candidates, which were filtered down to 4,400 (a 49% rejection rate). Filtering criteria included removing duplicates, validating technical accuracy, rephrasing for clarity, and excluding questions that drifted into adjacent domains (examples provided in Appendix A.2).
>
> For open-ended grading, responses were evaluated based on factual correctness and conceptual completeness. We acknowledge that manual grading limits scalability, which is why multiple-choice questions serve as the primary evaluation method while open-ended questions function as a supplementary diagnostic tool. The revision now documents these grading criteria.
>
> ---
>
> **W3. Human Baseline Scope**
>
> The 30-question subset includes questions from all seven benchmark topics with varying difficulty levels. We limited the survey to 30 questions out of respect for participants' time, as the technical difficulty of the questions requires sustained concentration. Extending the survey further would have risked lower completion rates and response quality.
>
> To strengthen the statistical significance of this baseline, **we have expanded the participant pool to 43 experts since submission.** These participants hold advanced degrees and possess active research experience in quantum computing. We have also added details on topic distribution and difficulty balance in the updated manuscript.

---

> ### Author Response · Authors · 2025-11-29
>
> **W4. Related Work Coverage**
>
> We have significantly expanded the Related Work section to discuss Qiskit HumanEval, QuanBench, QCircuitNet, and QASMBench. As the reviewer notes, these benchmarks target different goals from QC-Bench, and **we have revised the Related Work section to better contextualize our contribution alongside this existing work.**
>
> ---
>
> **W5. Data Release**
>
> The benchmark dataset and associated codes can be accessed through the Harvard Dataverse:
> [https://dataverse.harvard.edu/dataset.xhtml?persistentId=doi:10.7910/DVN/OG3VAR](https://dataverse.harvard.edu/dataset.xhtml?persistentId=doi:10.7910/DVN/OG3VAR)
>
> We will ensure a direct link is included in the manuscript once the review process is concluded.
>
> ---
>
> **W6. Writing Improvements**
>
> We have revised the abstract to improve readability while retaining the key findings, and corrected the Table 2 caption to "Accuracy above 90%."
>
> ---
>
> We sincerely thank the reviewer for the constructive feedback. The Related Work section is now much more comprehensive, the abstract is now more concise, and we have expanded the human baseline to 43 participants along with additional dataset documentation in Section 3. We would be happy to provide any additional clarifications.

---

### Official Review · Reviewer_xnJY · 2025-10-31

**Soundness:** 2
**Presentation:** 4
**Contribution:** 2
**Rating:** 4
**Confidence:** 3

**Summary:**

The authors introduce QC-Bench, the first LLM benchmark dataset for evaluating LLM knowledge of quantum computing. QC-Bench contains both human-authored and LLM-generated questions, using a variety of formats including multiple choice, true and false, and open-ended questions. The benchmark also includes 500 questions that have been translated into Spanish and French.

The authors evaluate a number of LLMs on QC-Bench, finding that newer, larger models outperform older and smaller models. Efforts are made to analyze LLM performance along a variety of axes, including question content.

The authors also show how to use QC-Bench for model fine-tuning.

**Strengths:**

1. Standard Benchmarking Approach: The paper checks many standard boxes for creating a benchmark, including the use of human and machine-generated questions grounded in the outputs of subject matter experts (SMEs).
2. Multilingual Question Set: The inclusion of a multilingual question set enhances the accessibility and applicability of the benchmark across different language speakers.
3. Diverse Question Formats: The benchmark incorporates open-form questions, allowing for a broader range of responses and insights into the capabilities of large language models (LLMs).
4. Domain-Specific Focus: The QC-Bench dataset is built on a review of over 200 peer-reviewed research papers, ensuring that the questions are relevant to the field of quantum computing and not just general science topics.
5. Human-Authored Evaluation Questions: The inclusion of 1200 human-authored evaluation questions adds credibility and relevance to the benchmark, as they are designed to reflect foundational questions and state-of-the-art research in quantum computing.

**Weaknesses:**

1. Lack of Novelty in Benchmarking: The paper fails to advance the science of benchmarking, offering no new approaches or methodologies for evaluating LLMs in the context of quantum computing.
2. Unclear Value Proposition: The benchmark raises questions about its actual value and whether it will encourage critical examination of LLM responses to quantum computing questions.
3. Limited Analysis Capabilities: The benchmark does not enable uncertainty quantification or more in-depth analysis of responses beyond traditional scoring methods, limiting its utility for comprehensive evaluation (cf. https://arxiv.org/pdf/2411.00640).
4. No Real-World Performance Correlation: There is a failure to tie performance on the benchmark to real-world performance or utility, which diminishes the relevance of the findings.
5. Insufficient Data for Conclusions: The conclusions regarding model performance on basic versus advanced topics lack supporting data on the distribution of these concepts within each question category, making it difficult to critically assess the results. Moreover, it is unclear if conclusions such as "they lack understanding of cutting-edge developments that define the current research frontier" are truly supported by models performing less well on quantum security questions (a relatively niche sub-field of quantum computing).
6. Questionable Relevance of Topics: The large focus on "quantum security" and lack of clear room for questions on noise characterization, quantum hardware, and quantum control questions whether the benchmark truly reflects the broader literature and raises concerns about the representativeness of the benchmark questions.
7. Lack of Human Subject Details: The paper does not provide summary statistics on the human participants, making it difficult to evaluate the expertise of the human subjects involved in the study.
8. Statistical Power Concerns: The administration of only 30 questions to 16 human subjects raises questions about the statistical power of the study and the reliability of the results.
9. Absence of Ethical Review Information: There are no details regarding the human subject review board process, which is a significant oversight given the use of human participants in the research.
10. Potential Bias in Grading: The paper does not address the possibility that human graders may have a generic preference for answers from larger models, regardless of the answers factual content, which could skew the evaluation results when comparing the performance of larger models to smaller models.

**Questions:**

Can you provide clarity on what each subject area (e.g., quantum security, gates & circuits) covers?

Can you provide more in-depth analysis supporting the claim that models perform better on well-established fundamentals and less well on cutting-edge research questions?

How can you enable a more statistically rigorous analysis of an LLM's capabilities using the QC-Bench (e.g., along the lines of https://arxiv.org/pdf/2411.00640)?

**Details Of Ethics Concerns:**

The study included 16 human survey respondents, but doesn't include any information about an ethical review process or, for example, institutional review board oversight.

---

> ### Author Response · Authors · 2025-11-29
>
> We thank the reviewer for the thorough evaluation and for acknowledging the depth of the benchmark, including the multilingual question sets, diverse question formats, and human-authored questions covering both foundational and state-of-the-art research. Our responses to the specific points raised follow.
>
> ---
>
> **Ethics Review**
>
> All participants were volunteers recruited from quantum computing research communities and professional networks. No compensation was provided, and no personal data was collected beyond standard survey demographics (education level, years of experience, and age group). **The study involved anonymous technical assessment with no sensitive data collection or potential harm to participants.** This follows standard practices for academic surveys in technical fields. We have added clarification regarding the human subject study in the updated manuscript.
>
> ---
>
> **W1. Lack of Novelty in Benchmarking**
>
> **The paper does not claim to advance the science of benchmarking.** Our contribution is a domain-specific benchmark for quantum computing, similar to other technically challenging benchmarks that introduce evaluation resources for underexplored areas. This is well within the scope of the venue. Existing quantum computing benchmarks such as Qiskit HumanEval, QuanBench, and QASMBench address code generation and circuit implementation, leaving conceptual knowledge evaluation unaddressed. Our contribution fills this gap.
>
> ---
>
> **W2. Unclear Value Proposition**
>
> **The benchmark reveals knowledge gaps across multiple topics**, as shown in our results. Models perform noticeably worse on certain advanced topics compared to foundational material, which we believe will encourage critical examination of LLM responses in these areas. Practitioners should be aware that models are less reliable on advanced and rapidly evolving topics than on well-established concepts, and our benchmark provides concrete evidence to guide this understanding.
>
> ---
>
> **W3. Limited Analysis Capabilities**
>
> **We have evaluated models from multiple angles to the extent feasible within the scope of this work:** multiple question formats (multiple-choice, true/false, open-ended), fine-tuning experiments on smaller models, evaluation of models with internet access using agentic and deep research modes, and multilingual evaluation across three languages. Additional metrics such as uncertainty quantification would be valuable, but given the scope of the paper and the breadth of evaluation already conducted, we leave this as a direction for future work.
>
> ---
>
> **W4. No Real-World Performance Correlation**
>
> **Ensuring that models provide accurate conceptual understanding of quantum computing has direct real-world utility.** Researchers, students, and practitioners increasingly use LLMs for learning, explaining theoretical concepts, and understanding technical papers. Benchmarking factual accuracy on these topics ensures that users can trust model outputs for educational and research purposes. Quantum computing is still at an early stage where foundational understanding matters significantly, and verifying that models have reliable knowledge in this domain is essential before they can be trusted for more applied tasks.
>
> ---
>
> **W5. Insufficient Data for Conclusions**
>
> **The claim reflects a consistent pattern across the benchmark.** Quantum security as a field has seen significant recent activity, and our questions cover material such as novel attack vectors and recently identified vulnerabilities. Models struggled with these questions because this material is specialized and less represented in general training corpora.

---

> ### Author Response · Authors · 2025-11-29
>
> **W6. Questionable Relevance of Topics**
>
> **The benchmark covers these topics across multiple categories.** Noise characterization and decoherence are addressed within "Error Correction," while quantum hardware and quantum control appear within "Gates & Circuits." There is some overlap between categories given the interconnected nature of these concepts. Quantum security receives substantial coverage because it represents a rapidly evolving area where models consistently struggle. We have clarified the scope of each category in Section 3.
>
> ---
>
> **W7. Lack of Human Subject Details**
>
> **Appendix A.5 provides detailed information on all participants**, including education level (BS/MS/PhD distribution), years of quantum computing experience, and age group.
>
> ---
>
> **W8. Statistical Power Concerns**
>
> **Since submission, we have expanded the participant pool from 16 to 43 researchers.** The 30-question sample was designed to balance comprehensive topic coverage (all seven categories represented) with realistic completion time (20-40 minutes). Extending beyond 30 questions would push completion time past an hour, which we considered too demanding for volunteers.
>
> ---
>
> **W9. Absence of Ethical Review Information**
>
> Addressed in the Ethics Review section above.
>
> ---
>
> **W10. Potential Bias in Grading**
>
> **For the 5,400 multiple-choice and 416 true/false questions, grading is fully automated based on exact answer matching**, eliminating any possibility of human bias. For the 421 open-ended questions, graders evaluated responses against predefined correct answers and were not informed which model produced which response. We acknowledge that subjective bias could still be present in open-ended evaluation, though this subset represents less than 7% of the total benchmark. We have added this clarification to Section 4.
>
> ---
>
> ## Response to Questions
>
> **Q1. Can you provide clarity on what each subject area (e.g., quantum security, gates & circuits) covers?**
>
> We have added detailed descriptions of each topic category in Section 3, noting that some concepts naturally appear across multiple categories given the interconnected nature of quantum computing.
>
> **Q2. Can you provide more in-depth analysis supporting the claim that models perform better on well-established fundamentals and less well on cutting-edge research questions?**
>
> Table 2 provides accuracy breakdowns by topic, showing consistent patterns across all models. We have expanded our discussion of these results in Section 5, detailing our observations regarding performance variations across topics.
>
> **Q3. How can you enable a more statistically rigorous analysis of an LLM's capabilities using the QC-Bench?**
>
> Adding error bars and confidence intervals would enhance the interpretability of the results. However, given the extensive nature of the evaluation (31 models, 6,237 questions, multiple formats, three languages, fine-tuning experiments, and agentic evaluation modes), incorporating this level of statistical analysis was not feasible within the current format. We have acknowledged this as a direction for future work in Section 6.
>
> ---
>
> Thank you very much for your review and constructive comments. In response to the concerns raised, we have expanded the human participant pool to 43 researchers, added detailed topic descriptions in Section 3, clarified the grading methodology in Section 4, and included discussion of future directions in Section 6. We would be happy to provide any additional clarifications.

---

### Official Review · Reviewer_bYba · 2025-10-31

**Soundness:** 2
**Presentation:** 3
**Contribution:** 1
**Rating:** 2
**Confidence:** 4

**Summary:**

This paper introduces QC-Bench, a comprehensive benchmark for evaluating LLM's knowledge of quantum computing. The benchmark contains 6,237 questions across seven core topics of quantum computing and are collected from over 200 peer-reviewed papers. The authors evaluate more than 30 frontier LLMs and compare their performance against human experts.

**Strengths:**

1. I appreciate effort of including human quantum experts in both data curation and evaluation processes.

2. The benchmark does a good job in balancing both fundamental quantum theory and more recent research.

3. The paper is well-written.

**Weaknesses:**

1. I think the paper fails to convince the reader the significance of the benchmark. First, it's unclear whether QC-Bench evaluates simple quantum knowledge retrieval or reasoning like a quantum scientist. Will the modern LLMs with web search abilities solve these questions by looking up information online? If yes, I am not sure that the benchmark will stand test of time. More critically, the paper does not convince me that "acing in this benchmark will mean the model will be a good quantum scientist".

2. The memorization analysis is inadequate. The authors claim to address memorization concerns by testing if the model can reproduce the answers verbatim, but this approach has serious problems. it does not test whether models have memorized the concepts and facts from source papers, only whether they've memorized exact phrasings. Given that many papers are quite old and famous, it's pretty safe to assume they indeed apper in the training set.

3. The multilingual evaluation lacks clear motivation. The multilingual experiments present French and Spanish translations of 500 questions, but it's unclear what relevant research question this addresses. If anything, this might show that LLMs might be memorizing the source materials.

4. The fine-tuning experiments feel underdeveloped. Section 4.4 presents fine-tuning results on 5 models but offers limited analysis. Why do some models improve (Llama-3.1-8B: +5%) while others decline (gpt-j-6b: -7%)? The conclusions offered in the paper is not satisfying.

**Questions:**

1. Will LLM ace QC-bench by doing internet search?

2. Can you provide analysis showing whether model performance correlates with source paper citation counts or publication dates?

3. What hypothesis were you testing with the French and Spanish translations?

4. Do you think the performance decrease in a differnet language implies memorization?

---

> ### Author Response · Authors · 2025-11-29
>
> We thank the reviewer for recognizing the inclusion of human quantum experts in data curation and evaluation, the balance between fundamental theory and recent research, and the writing quality of the paper. We address each concern below.
>
> ---
>
> **W1. Significance of the Benchmark**
>
> **QC-Bench evaluates reasoning, not simple retrieval.** The questions require expert-level understanding of quantum computing concepts, not just the ability to look up facts. To directly test whether web search would trivialize the benchmark, we added a new subsection (Section 4.5, highlighted in blue) evaluating models with Agentic and Deep Research modes. These models have full internet access and extended time to research concepts, yet overall performance improved by an average of only 0.84 percentage points. Models still struggled with certain concepts even with web access, confirming that the benchmark tests genuine reasoning rather than retrievable knowledge.
>
> This directly addresses the concern about standing the test of time. Other technically demanding benchmarks like Humanity's Last Exam, MMLU, and SWE-Bench have remained valuable precisely because they test challenging domain knowledge that cannot be trivially solved. QC-Bench follows this approach, and our results with web-enabled models support its longevity. Full results are included in our response to Question 1 below.
>
> Regarding the "good quantum scientist" characterization, this is not our claim. Strong benchmark performance demonstrates a model's reliability in quantum computing contexts, which is what benchmarks are designed to measure.
>
> ---
>
> **W2. Memorization Analysis**
>
> **We did not claim to address memorization by testing verbatim reproduction. What we ensured is that questions themselves are not reproduced verbatim from source papers.** For questions generated with LLM assistance, we verified that questions were reformulated rather than copied directly. This prevents models from simply retrieving exact question-answer pairs from training data.
>
> Additionally, QC1000, a substantial portion of our benchmark, was generated entirely by researchers without any LLM assistance. Performance patterns across models, particularly the consistent decline on advanced topics like quantum security, suggest that models are not simply recalling memorized content.
>
> ---
>
> **W3. Multilingual Evaluation Motivation**
>
> Scientific and technical work happens globally, yet evaluation of AI systems for technical domains remains predominantly English-focused. Understanding how these systems perform across languages matters for researchers and practitioners working in their native languages. Our multilingual evaluation helps address this gap and enables assessment of how quantum computing knowledge transfers across languages.
>
> ---
>
> **W4. Fine-Tuning Experiments**
>
> The purpose of Section 4.4 was to demonstrate the benchmark's utility for model adaptation, not to conduct exhaustive fine-tuning analysis. **The inconsistent results reflect inherent instability in smaller models during domain-specific adaptation. Fine-tuning larger models that already achieve strong performance remains technically infeasible, which limits the scope of such experiments.** Given these constraints, we added Section 4.5 evaluating agentic and deep research capabilities as a more practical direction for enhancing performance on technical benchmarks.

---

> ### Author Response · Authors · 2025-11-29
> **Response to Questions**
>
> **Q1. Will LLMs ace QC-Bench by doing internet search?**
>
> We tested this directly by evaluating models with Agentic and Deep Research modes, where models have full internet access and extended time to research concepts. Results are shown below:
>
> | Model | Before (%) | After (%) | Improvement |
> |-------|------------|-----------|-------------|
> | Claude Sonnet 4.5 (Research Mode) | 91.80 | 92.20 | +0.40 |
> | Claude Sonnet 4.5 (Extended Thinking) | 91.80 | 92.60 | +0.80 |
> | GPT-5.1 (Deep Research) | 91.40 | 92.80 | +1.40 |
> | GPT-5.1 (Agent Mode) | 91.40 | 91.20 | -0.20 |
> | Gemini 3 (Deep Research) | 87.40 | 89.20 | **+1.80** |
>
> Models showed an average improvement of only 0.84 percentage points across 500 questions. While internet access provides some benefit for general knowledge retrieval, heavily technical domains like quantum computing appear to remain challenging even with web search capabilities. These findings are detailed in the new Section 4.5 (highlighted in blue in the updated manuscript).
>
> **Q2. Can you provide analysis showing whether model performance correlates with source paper citation counts or publication dates?**
>
> Due to the scale of our dataset and the number of models evaluated, we were unfortunately not able to include this analysis. The benchmark questions are reformulated concepts from source papers rather than direct extractions, which adds complexity to such correlation analysis. This remains an interesting direction for future work.
>
> **Q3. What hypothesis were you testing with the French and Spanish translations?**
>
> We tested whether quantum computing knowledge transfers uniformly across languages or remains language-dependent. Given that most training data for these models is in English, we wanted to understand how well quantum concepts are represented and accessible in other languages. We have clarified this motivation in the updated manuscript.
>
> **Q4. Do you think the performance decrease in different languages implies memorization?**
>
> Memorization in language models remains an open research question. That said, **we believe memorization likely contributes to some extent, since most quantum computing research is published in English, giving models direct exposure to these concepts without translation.** Other factors may also be at play, including differences in how quantum terminology is standardized across languages and the availability of technical content in multilingual training data.
>
> ---
>
> We sincerely thank the reviewer for the detailed feedback, which has substantially improved the quality of the paper. In response to this review, we added a new subsection (Section 4.5) with comprehensive evaluation of agentic and deep research capabilities, clarified our hypotheses for the multilingual evaluation, and strengthened the discussion of fine-tuning limitations. We believe these revisions address the concerns raised and would be happy to provide any additional clarifications.

---

### Author Response · Authors · 2025-12-04
**Summary of Revisions**

Dear Area Chairs, Senior Area Chairs, Program Chairs, and Reviewers,

We want to thank the reviewers for their feedback. As the rebuttal period was cut short
due to technical issues at the conference, we were unable to engage in discussion with
the reviewers about the paper. We provide here a summary of the main points raised and
our response to them.

The reviewers raised four main issues: whether the benchmark could be solved with web
search, whether our human baseline was robust enough, how the work relates to existing
quantum computing benchmarks, and concerns about potential bias from LLM-extracted
questions.

We've addressed all of these. First, we ran experiments (**Section 4.5**) with
models that have internet access and extended reasoning time. These models improved by
less than 1 percentage point on average, showing that web search doesn't trivialize the
benchmark. Second, we grew our human expert pool from **16 to 43 participants**, all
with education and research experience in quantum computing. Third, we expanded the
Related Work section to properly position our contribution alongside existing benchmarks
like Qiskit HumanEval and QuanBench. Fourth, we clarified the dataset construction
process: **QC1000 is entirely human-authored**, and the remaining 4,400 questions went
through extensive filtering (49% rejection rate) with manual validation of technical
accuracy. The reviewers questioned whether high performance on foundational topics
limits the benchmark's value, but the results show persistent gaps on advanced material
and no model breaking 90% overall on the full QC5400 benchmark.

Beyond these main changes, we added detailed descriptions of each topic category in
Section 3, documented the grading methodology for open-ended questions in Section 4,
clarified the ethics review process, and revised the abstract for better readability.

We thank the reviewers for acknowledging the depth of the benchmark and the
comprehensive evaluation approach, and for their constructive suggestions, which have
been incorporated into the revised paper.

---

### Note · Program_Chairs · 2026-01-17
**Submission Desk Rejected by Program Chairs**

The following references in this submission do not refer to real documents and/or have major errors in bibliographic information:

 David Avis, Charles Jordan, Jun Imoto, Yuki Sasaki, Sven Thomassen, Taiga Tsuda, and Seiya Yamanaka. Comparing small-and large-scale quantum computers using circuit simulation. arXiv preprint arXiv:1904.11502, 2019.